# Clinical Evidence of Bee Venom Acupuncture for Ankle Pain: A Review of Clinical Research

**DOI:** 10.3390/toxins17050257

**Published:** 2025-05-21

**Authors:** Soo-Hyun Sung, Hyein Jeong, Jong-Hyun Park, Minjung Park, Gihyun Lee

**Affiliations:** 1Department of Policy Development, National Institute of Korean Medicine Development, Seoul 04554, Republic of Korea; koyote10010@nikom.or.kr; 2Department of Preventive Medicine, College of Korean Medicine, Kyung Hee University, Seoul 02447, Republic of Korea; frogcream@gmail.com; 3Department of Pathology, College of Korean Medicine, Daegu Haany University, 1 Haanydaero, Gyeongsan 38610, Republic of Korea; moguri@dhu.ac.kr; 4Department of Preventive Medicine, College of Korean Medicine, Gachon University, Seungnam 13120, Republic of Korea; 5College of Korean Medicine, Dongshin University, Naju 58245, Republic of Korea

**Keywords:** bee venom, apitherapy, acupuncture, ankle function

## Abstract

The prevalence of ankle pain in adults is 9–15%, with up to 45% of sports-related injuries attributed to ankle pain and injuries. If ankle pain is not controlled in a timely manner, it can lead to ankle instability, resulting in further damage, recurrence of pain, and secondary injuries. The present study aimed to assess the therapeutic potential and safety profile of bee venom acupuncture (BVA) in the management of ankle pain. Ten electronic databases were searched for articles published up to March 2025. We included clinical studies that utilized BVA for the treatment of ankle pain and studies that included pain- and function-related assessment tools. The safety of bee venom acupuncture (BVA) was assessed by extracting adverse events from the included studies and categorizing them according to the Common Terminology Criteria for Adverse Events (CTCAE). A total of 14 clinical studies were selected, of which 9 were case reports, 2 were case-controlled clinical trials (CCTs), and 3 were randomized controlled trials (RCTs). The conditions causing ankle pain were mostly traumatic (42.9%), followed by inflammatory (21.4%) and neuropathic disorders (14.3%). BVA was applied at concentrations ranging from 0.05 to 0.5 mg/mL, with a per-session volume ranging from 0.04 to 2.5 mL. In most studies, BVA was reported to improve both ankle pain and function simultaneously. Among the 14 studies, four participants reported adverse events following BVA treatment, all of which were classified as grade 1 or grade 2, indicating mild to moderate severity. This review suggests that BVA may be recommended for controlling ankle pain based on clinical evidence. However, the number of high-quality RCTs is limited, and half of the studies did not report side effects, indicating the need for further clinical research to verify its safety and efficacy.

## 1. Introduction

Ankle pain is often accompanied by injuries such as sprains, strains, and contusions [1,2]. Ankle pain affects approximately 11.7% [3] of community-dwelling adults aged 50 and over, while foot and ankle pain combined affect about 20% of middle-aged and older adults [4]. Ankle pain is generally classified as traumatic or non-traumatic in origin. Non-traumatic causes include neuropathic or inflammatory conditions, while traumatic incidents account for a significant proportion [5]. Specifically, up to 45% of sports-related injuries are associated with ankle issues [6,7].

Ankle injuries and pain require immediate attention and care, as delayed treatment can lead to persistent symptoms, recurrent issues, functional limitations, mood disturbances, and diminished overall well-being [6]. Chronic ankle injuries and pain contribute to ankle instability, which is a major factor in the recurrence of damage and pain [8]. Conventional medicine approaches for treating ankle pain include the RICE protocol (rest, ice, compression, and elevation), non-steroidal anti-inflammatory drugs (NSAIDs), immobilization (e.g., leg cast), functional support (e.g., ankle brace), exercise, manual mobilization, and surgical therapy, with complementary therapies such as acupuncture also being recommended [9,10,11].

Bee venom (BV) is a complex secretion from honeybee glands, consisting of various bioactive components, including peptides such as melittin, apamin, and adolapin; enzymes such as phospholipase A_2_ and hyaluronidase; as well as amino acids and volatile compounds [12,13]. In animal studies, BV has been shown to have anti-inflammatory, antinociceptive, anticancer, and anti-allergic effects [13,14,15]. However, since BV is an animal toxin, clinical evidence is needed to reduce side effects and ensure its effectiveness for clinical use. In some East Asian countries, traditional clinics use bee venom acupuncture (BVA) for disease treatment [16]. BVA is a therapeutic technique that involves the subcutaneous injection of purified and diluted BV into specific acupuncture points [17].

Upon reviewing existing literature on BVA, numerous studies have explored its application in various conditions, including Parkinson’s disease, shoulder pain, bone fractures, neck pain, lower back pain, and rheumatoid arthritis (Appendix A [18,19,20,21,22,23,24,25,26,27]). However, there appears to be a lack of studies specifically addressing the efficacy of BVA for ankle pain. Therefore, the authors aim to review clinical studies on the use of BVA for ankle pain in order to evaluate its effectiveness and safety.

## 2. Results

### 2.1. Study Description

A total of 14 studies [28,29,30,31,32,33,34,35,36,37,38,39,40,41] were included in this review based on the predefined inclusion criteria (Figure 1). The first study in South Korea using BVA for ankle pain was published in 2002 (Figure 2). From 2001 to 2024, a minimum of zero and a maximum of three studies were published annually. No relevant studies were published in 2001, 2007, 2010, 2012, 2013, or from 2018 to 2024. This review includes nine case reports, two case-controlled clinical trials (CCTs), and three randomized controlled clinical trials (RCTs). All 14 studies were conducted in South Korea. The characteristics of the 14 included studies are summarized in Table 1.

### 2.2. Medical Conditions and Sample Size

A total of 10 medical conditions were identified across 14 studies. These conditions were categorized into four groups: traumatic, neuropathic, inflammatory, and other conditions. Traumatic conditions, including ankle sprain, traumatic partial tear, and malleolus avulsion fracture, were the most frequently studied, accounting for six studies (42.9%). Neuropathic conditions, such as tarsal tunnel syndrome and peroneal nerve palsy with foot drop, were investigated in two studies (14.3%). Inflammatory conditions, including synovitis of the ankle joint with osteonecrosis, rheumatoid arthritis, and acute inflammatory arthritis, were examined in three studies (21.4%). Finally, other conditions, including post-operative cases and anterior impingement syndrome, were reported in two studies (14.3%). One study [28] did not report the underlying medical conditions associated with ankle pain. The number of studies for each condition, along with the average and range of patient numbers, is presented in Table 2.

The analysis included data from 14 articles, encompassing a total of 112 patients experiencing ankle pain. The number of participants in each study varied, with sample sizes ranging from 1 to 32.

### 2.3. BVA Treatment

BVA treatment was administered through injections into acupoints using a syringe, with concentrations varying based on the disease category. For traumatic conditions (ankle sprain, traumatic partial tear, malleolus avulsion fracture), the BVA concentration ranged from 0.05 to 0.5 mg/mL, with a volume per session ranging from 0.04 to 2.5 mL and a total volume of 0.2 to 16.0 mL. The concentrations and volumes for other disease categories, including neuropathic, inflammatory, and other conditions, are summarized in Table 3. Additionally, concentration data were not reported for one case (rheumatoid arthritis [35]), volume per session was missing for one case (peroneal nerve palsy with foot drop [40]), and total volume was not reported in one case (peroneal nerve palsy with foot drop [40]). Details regarding the bee species, the composition of bee venom, and its method of preparation are available in Appendix A [42].

### 2.4. Outcome Measures

A total of nine outcomes were observed in 14 clinical studies. The results for each outcome measure were categorized as “statistically significant improvement”, “improvement”, or “no improvement” (Figure 3). The visual analog scale (VAS) was the most commonly employed instrument for evaluating the severity of ankle pain. The numeric rating scale (NRS), a measure similar to VAS, was also widely utilized. Notably, no respondents in the VAS and NRS assessments reported “not improved”; all responses indicated either “improved” or “statistically improved”. Additionally, several studies assessed the range of motion (ROM) of the ankle. While most responses indicated improvement, there was one case where the outcome was classified as “not improved”.

### 2.5. Adverse Events

Among the 14 studies, excluding the 7 clinical studies categorized as ‘not reported’, 7 studies were included in the analysis (Figure 4). Of these, three studies reported no adverse events, while the remaining four studies documented a total of five different adverse effects. Mild to moderate adverse reactions, such as itching (*n* = 2) and redness (*n* = 2), were reported. Additionally, other symptoms, including local edema (*n* = 1), moderate joint irritation (*n* = 1), and mild muscle pain with chilling (*n* = 1), were observed. According to the Common Terminology Criteria for Adverse Events (CTCAE), adverse events in four patients were classified as grade 1 in three cases and grade 2 in one case.

### 2.6. Risk-of-Bias Assessment (ROB)

A Cochrane risk-of-bias (ROB) assessment was conducted for three RCTs (Table 4) [31,34,36]. All three studies were assessed as having a low risk of selection bias. One study [31], which compared BVA with saline injection, was rated as having adequate blinding of participants. However, the other two studies [34,36], which compared BVA with acupuncture, were assessed as having a high risk of performance bias. None of the three studies [31,34,36] provided information on blinding of outcome assessors. Two studies [34,36] had dropout rates exceeding 20%, leading to a high risk of attrition bias. Finally, all three studies [31,34,36] did not report study protocol registration, resulting in an unclear risk of bias in this domain.

### 2.7. Co-Interventions

Among the 14 studies analyzed, 11 studies utilized co-interventions. The data presented in Figure 5 represent the frequency of co-interventions used in these 11 studies. Acupuncture was the most commonly used co-intervention (*n* = 9), followed by herbal medicine (*n* = 6) and physical therapy (*n* = 3). Other interventions, including infrared therapy (*n* = 2), moxibustion (*n* = 2), and cupping therapy (*n* = 2), were reported less frequently. Additionally, ankle support (*n* = 1), electro-acupuncture (*n* = 1), and warm acupuncture (*n* = 1) were the least frequently used interventions.

## 3. Discussion

Among the included studies, 42.9% (*n* = 6) reported ankle pain due to trauma. Traumatic ankle injuries, such as sprains, are highly prevalent in the general population and occur frequently among athletes [43]. Athletes often have to play despite ankle injuries and pain, which can result in joint instability and increase the risk of recurrence, potentially causing chronic conditions [44]. Therefore, management programs for returning to sport after an ankle injury are crucial. In this study, BVA showed improvement not only in pain assessment tools for the ankle but also in functional evaluation tools. This suggests that BVA may be recommended as part of an ankle injury and pain management program for athletes. Management programs should be personalized, combining taping, physical therapy, medication, and other interventions based on the athlete’s condition. Future research is needed to evaluate the effectiveness of combining BVA with existing treatments for ankle injury and pain management.

BVA has also shown improvements in ankle pain caused by nerve compression and paralysis. Treatments for nerve compression and paralysis in the ankle typically involve physical therapy, nerve block medications, weight management, and, in severe cases, surgery [45]. BVA has been proven effective in animal models for nerve-related pain and is known to promote recovery and regeneration of damaged nerves [14]. Therefore, BVA may be recommended for managing nerve-related ankle pain. Future research should focus on developing guidelines for the combination of BVA with other treatments and determining the optimal stage of treatment prior to considering surgical interventions.

The main component of BV, melittin, possesses both anti-inflammatory and analgesic effects. Melittin acts on the nerves to suppress pain signals and alleviates inflammation, which reduces pain and improves function [46,47]. However, melittin also exhibits cytotoxicity against normal cells, leading to organelle degeneration and necrosis, and demonstrates potent hemolytic activity [48,49]. Phospholipase A_2_ has been reported to exert various therapeutic effects, including analgesic, wound healing, anticancer, antiviral, antibacterial, antiparasitic, and anti-angiogenic activities; however, it has also been shown to be neurotoxic to neurons and glial cells [50,51]. Apamin, a neurotoxic small peptide, irreversibly blocks small-conductance calcium-activated potassium channels, thereby contributing to its antinociceptive effects [52]. Nonetheless, apamin has been associated with adverse effects such as ataxia and tremors, and in animal models, its administration has resulted in pulmonary hyperinflation, petechial hemorrhages, and congestion in the liver, spleen, and kidneys [53]. Given that BV possesses both therapeutic benefits and toxicological risks, careful monitoring of the toxicity associated with BVA is warranted.

In the study by Liu et al. [54], the toxicological effects of BV were evaluated in rats following administration of high (120 mg/kg), medium (60 mg/kg), and low (8 mg/kg) doses. Severe toxicity was observed at the high dose, while the medium dose resulted in changes in mental status, appetite, and respiration. No significant toxic effects were detected at the low dose. Future in vivo and in vitro studies are warranted to identify the optimal dose that minimizes toxicity while maintaining therapeutic efficacy for ankle pain. In the clinical setting, analysis of real-world data from Traditional Korean Medicine (TKM) clinics is also needed to inform the development of evidence-based guidelines for the use of BVA in the treatment of ankle pain.

Previous studies have reported on the adverse effects of different types of BV therapy (BVA, BV immunotherapy, and live bee sting) [55]. It was found that BVA had a higher proportion of no or mild-to-moderate adverse effects compared to BV immunotherapy and live bee stings. In contrast, the proportion of severe reactions was highest in BV immunotherapy (23%), followed by live bee sting (12.5%) and BVA (4.4%). No severe systemic reactions were observed in any of the three BV therapy types. This suggests that the treatment effects and adverse effects may vary depending on the type of BV therapy used clinically. Of the seven studies that reported BVA-related adverse effects in this research, three studies reported no adverse effects, while the remaining four studies reported only mild-to-moderate reactions. Additionally, in South Korea, BVA is administered after confirming BV hypersensitivity through a skin test and is applied at acupuncture points [17]. For BVA to be recommended as a treatment for ankle pain, further clinical studies with higher levels of evidence, such as RCTs, are needed.

Among the 14 included studies, 5 studies [28,29,30,31,33] utilized BV that was collected through electrical stimulation of live bees, followed by processing, drying, and dilution with normal saline. Unrefined BV may contain various impurities, which can lead to unexpected side effects, immune responses, and inconsistent therapeutic outcomes. In the study by Song et al. [31], it was unclear whether the reported case of “severe itching” was due to the lack of purification, an adverse effect of the active components of BVA, or an idiosyncratic patient reaction. In contrast, eight studies [32,34,36,37,38,39,40,41] used bee venom dry powder registered with the Korea Food and Drug Administration (FDA), which underwent sterilization, dilution with saline, and aseptic sealing during preparation [42]. While the adverse events reported in these eight studies were either mild or absent, the limited number of studies included in this review highlights the need for further research.

This study has several limitations. First, among the 15 studies, only 3 were high-quality RCTs [31,34,36]. RCTs are prospective studies that strictly control clinical research environments to measure and compare the therapeutic effects between experimental and control groups [56]. Furthermore, analyzing only those participants who completed the treatment as planned according to a defined treatment protocol is considered the most reliable and least biased approach [49]. Therefore, future studies should focus on conducting high-quality RCTs, and systematic reviews and meta-analyses derived from such trials. Secondly, all the studies considered in this analysis were carried out in South Korea. This implies the possibility of publication bias, and careful interpretation of the results is needed. Pharmacopuncture is utilized in 29.9% of TKM clinics in South Korea, with BVA identified as the most commonly applied treatment method among pharmacopunctures [17,57]. Therefore, future multinational, multicenter RCTs are necessary to verify the efficacy and safety of BVA. Thirdly, only half of the 14 studies reported adverse effects. In the seven studies that mentioned adverse effects, none reported serious adverse events; however, the evidence remains limited. Future clinical studies on BVA for ankle pain should include reporting on adverse effects.

RCTs, with their high level of evidence, play a crucial role in determining whether a treatment should be incorporated into clinical practice guidelines [56]. However, RCTs are time-consuming and expensive, and BVA, primarily used in East Asian countries, likely has a limited number of clinical studies. In such cases, establishing a real-world data collection platform based on primary healthcare institutions can help reduce bias and verify the safety and efficacy of BVA for treating ankle pain [58].

## 4. Conclusions

Findings from studies on BVA for ankle pain suggest its potential as a therapeutic option in clinical settings for managing traumatic, neuropathic, and inflammatory ankle pain. Since BVA involves the dilution of BV in saline, it may help reduce toxicity and achieve analgesic effects safely. However, among the 14 included studies, only 3 were randomized controlled trials (RCTs), and of these, only 1 was assessed as having high methodological quality. To establish the efficacy and safety of BVA for ankle pain, future large-scale, multicenter, and multinational RCTs are warranted. Furthermore, studies are needed to identify the optimal dosage that maximizes therapeutic effects while minimizing toxicity, with the ultimate goal of developing evidence-based treatment guidelines for ankle pain.

## 5. Materials and Methods

### 5.1. Data Sources and Searches

A comprehensive search was conducted across electronic databases, including international databases (PubMed, EMBASE, Cochrane Central Register of Controlled Trials, and CINAHL Plus) and Korean databases (KM base, RISS, National Library of Korea, ScienceON, OASIS, and the Korean Traditional Knowledge Portal). Relevant papers published up until March 2025 were included in the review.

The following search terms were used according to the language of each database: “bee venom OR bee toxin OR apitherapy OR bee venom therapy OR bee venom acupuncture” AND “ankle pain” AND “randomized clinical studies OR clinical studies OR randomized clinical trial OR case-controlled studies OR case-controlled trials OR case reports OR case series OR case studies”.

### 5.2. Study Selection

Two independent reviewers (H.I.J. and S.-H.S.) assessed the titles and abstracts of the retrieved articles for inclusion. In the second step, the full texts of the selected studies were examined to determine whether they met the inclusion criteria. All types of clinical studies were included, encompassing case reports, non-randomized controlled trials, and randomized controlled trials that assessed the efficacy and safety of BVA for ankle pain. Non-clinical studies, such as reviews, experimental studies, and surveys, were excluded. We included articles involving patients with ankle pain and did not impose any restrictions on participants’ sex, age, ethnicity, race, or the type of control group. Studies that included pain-related assessment tools (e.g., VAS, NRS) were included in the review. Disagreements were addressed and resolved through consultation with the corresponding authors (M.P. and G.L.).

### 5.3. Data Extraction

Data were independently extracted by two reviewers (H.I.J. and S.J.) using a pre-specified form. The extracted data included the first author, publication year, study design, patient characteristics, BVA intervention, outcome measures and results, side effects, and co-interventions. Any discrepancies were resolved after consultation with the corresponding authors (M.P. and G.L.).

### 5.4. Effectiveness and Safety Assessment

The effectiveness of BVA was evaluated using outcome measures applied in the included clinical studies. Pain scores (e.g., VAS, NRS, or symptom changes) and functional scores (e.g., ROM, American Orthopaedic Foot & Ankle Society Hindfoot Score (AHS), or walking tests) were considered the primary outcomes. Safety was assessed by collecting data on adverse events that occurred in the BVA group during the study period of each included clinical trial. The collected adverse events were classified according to the CTCAE [59] as follows: (1) Mild: asymptomatic or mild symptoms; clinical or diagnostic observations only; intervention not indicated; (2) Moderate: minimal, local, or noninvasive intervention (e.g., packing) indicated; limiting age-appropriate instrumental activities of daily living; (3) Severe or medically significant but not immediately life-threatening: hospitalization or prolongation of hospitalization indicated; disabling; limiting self-care activities of daily living; (4) Life-threatening consequences: urgent intervention indicated; (5) Death related to adverse event.

### 5.5. Quality Assessment of RCTs

For studies that included randomized controlled trials (RCTs), the quality assessment was conducted using the Cochrane risk-of-bias (ROB) tool [60]. The ROB tool consists of six domains: (1) sequence generation, (2) allocation concealment, (3) blinding of participants and personnel, (4) blinding of outcome assessment, (5) incomplete outcome data, and (6) selective outcome reporting. Each domain was rated as having a low, unclear, or high risk of bias.

## Figures and Tables

**Figure 1 toxins-17-00257-f001:**
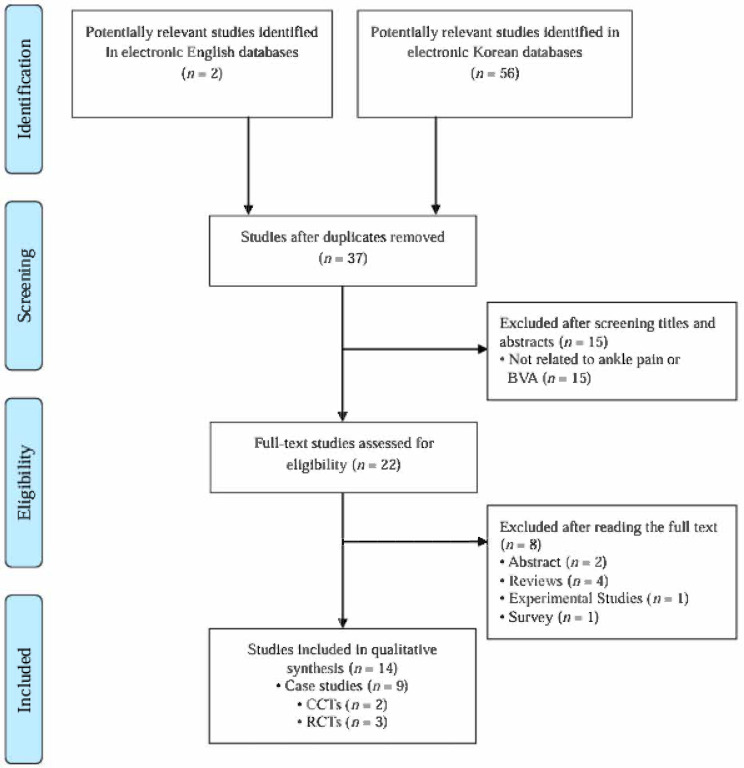
Overview of the study selection procedure according to the inclusion and exclusion criteria. BVA: bee venom acupuncture; CCTs: case-controlled clinical trials; RCTs: randomized controlled clinical trials.

**Figure 2 toxins-17-00257-f002:**
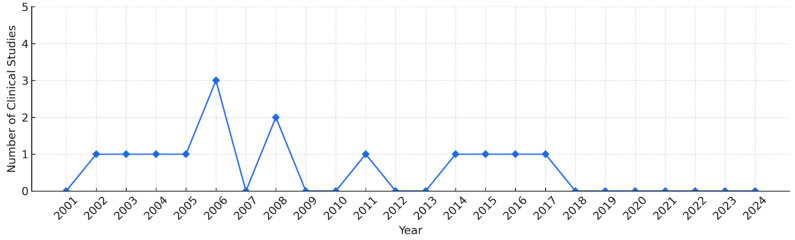
Yearly trends in the number of clinical trials of bee venom acupuncture in the treatment of ankle pain. Line graph showing the number of clinical studies published annually from 2001 to 2024.

**Figure 3 toxins-17-00257-f003:**
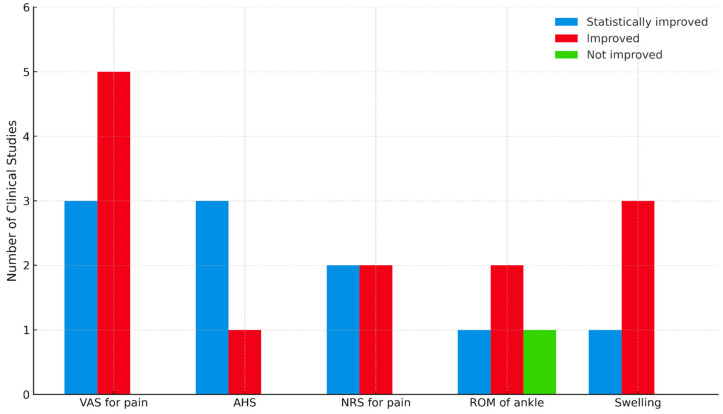
Clinical outcomes of bee venom acupuncture for the management of ankle pain. Only outcomes assessed in three or more studies were included. AHS: American Orthopaedic Foot & Ankle Society Hindfoot Score; NRS: numeral rating scale; ROM: range of motion; VAS: visual analog scale.

**Figure 4 toxins-17-00257-f004:**
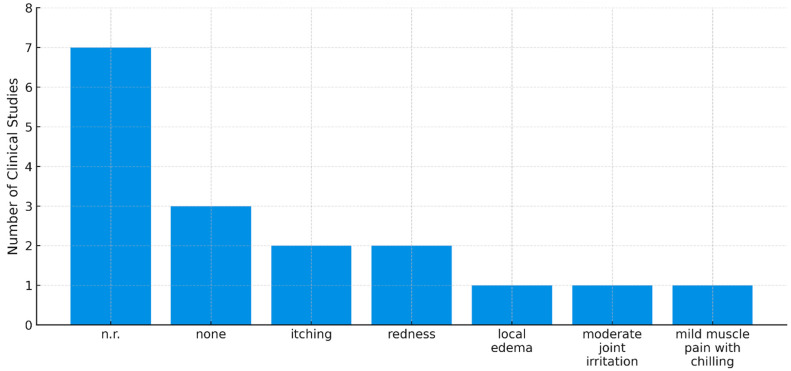
Adverse events associated with bee venom acupuncture in clinical studies for ankle pain treatment. Note: n.r. refers to studies where adverse events were not reported at all, whereas none indicates that studies explicitly stated no adverse events occurred; n.r.: not reported.

**Figure 5 toxins-17-00257-f005:**
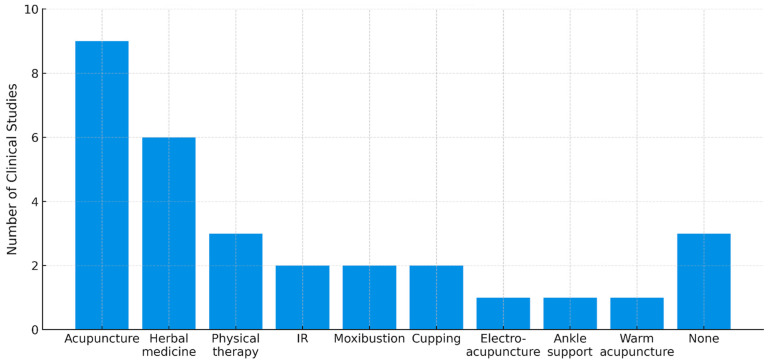
Co-intervention modalities in clinical trials for the treatment of ankle pain. IR: infrared radiation.

**Table 1 toxins-17-00257-t001:** Characteristics of included clinical studies.

	First Author (Year)	Study Design	Number of Patients	Diseases	Concentration and Volume	Outcome Measure	Main Result *	Adverse Events	Co-Intervention
1	Ahn [28] (2002)	Case report	*n* = 32	Patients with ankle pain	1. Concentration: 0.33 mg/mL 2. 1 session: 0.5 mL 3. Total 1 session: n.r. (0.1 cc/point, n.r.)	1. Symptom Changes	1. Improved	n.r.	1. Acupuncture 2. Herbal medicine 3. Cupping
2	Ryu [29] (2003)	Case report	*n* = 1	Synovitis of ankle joint with osteonecrosis of talus patient with ankle pain	1. Concentration: 0.05 mg/mL 2. 1 session: 0.1–2.5 mL 3. Total 15 sessions: 13.41 mL	1. Symptom changes (1) Pain (2) Swelling	1. (1) Improved (2) Improved	Moderate joint irritation, mild muscle pain with chilling in 1 case	None
3	Lee [30] (2004)	CCT	*n* = 16	Acute ankle sprain patients with ankle pain	1. Concentration: 0.05 mg/mL 2. 1 session: 0.1–2 mL 3. Total 4 sessions: 0.4–8 mL	1. VAS for pain 2. ROM of ankle 3. Swelling	1. *p* < 0.01 2. *p* < 0.01 3. *p* < 0.01	n.r.	None
4	Song [31] (2005)	RCT	*n* = 11	Acute ankle sprain patients with ankle pain	1. Concentration: 0.33 mg/mL 2. 1 session: 0.04 mL 3. Total 7 sessions: 0.28 mL	1. VAS for pain 2. AHS	1. *p* < 0.01 2. *p* < 0.001	Severe itching in 1 case	None
5	Choi [32] (2006)	Case report	*n* = 1	Post-operative patient with ankle pain	1. Concentration: 0.1 mg/mL, 0.25 mg/mL, or 0.5 mg/mL 2. 1 session: 0.1–0.4 mL (0.1 mg/mL), 0.2–0.4 mL (0.25 mg/mL), or 0. 2 mL (0.5 mg/mL) 3. Total 11 sessions: 1.0 mL (0.1 mg/mL), 1.6 mL (0.25 mg/mL), and 0.2 mL (0.5 mg/mL)	1. VAS for pain	1. Improved (10 to 2)	Local edema, redness in 1 case	1. Acupuncture 2. Herbal medicine
6	Kim [33] (2006)	CCT	*n* = 16	Ankle sprain patients with ankle pain	1. Concentration: 0.3 mg/mL 2. 1 session: 0.45 mL 3. More than 3 times in total: more than 1.35 mL	VAS for pain	*p* < 0.05	None	1. Acupuncture
7	Seo [34] (2006)	RCT	*n* = 11	Acute ankle sprain patients with ankle pain	1. Concentration: 0.25 mg/mL or 0.1 mg/mL 2. 1 session: 0.1–0.3 mL 3. Total 3 sessions: 0.3–0.9 mL	1. NRS for pain 2. AHS	1. *p* < 0.01 2. *p* < 0.01	n.r.	1. Acupuncture 2. Physical therapy 3. IR 4. Ankle support
8	Choi [35] (2008)	Case report	*n* = 1	Rheumatoid arthritis patient with ankle pain	1. Concentration: n.r. 2. 1 session: 1.0 mL 3. Total 16 sessions: 16.0 mL	1. VAS 2. Walking	1. Improved (10 to 2) 2. Improved (impossible to possible)	n.r.	1. Acupuncture 2. Herbal medicine
9	Kang [36] (2008)	RCT	*n* = 18	Acute ankle sprain patients with ankle pain	1. Concentration: 0.125 mg/mL 2. 1 session: 0.6 mL 3. Total 3 sessions: 1.8 mL	1. NRS for pain 2. AHS	1. *p* < 0.001 2. *p* < 0.001	None	IR
10	Park [37] (2011)	Case report	*n* = 1	Anterior impingement syndrome of ankle patient with ankle pain	1. Concentration: 0.1 mg/mL 2. 1 session: 0.06 mL 3. Total 5 sessions: 0.3 mL	1. VAS for pain 2. Intensity score (1) Tenderness (2) Swelling	1. Improved (6 to 1) 2. (1) Improved (2 to 0) (2) Improved (2 to 0)	n.r	1. Acupuncture 2. Physical therapy
11	Won [38] (2014)	Case report	*n* = 1	Tarsal tunnel syndrome patient with ankle pain	1. Concentration: 0.1 mg/mL or 0.175 mg/mL 2. 1 session: 0.2–0.6 mL (0.1 mg/mL) or 0.4 mL (0.175 mg/mL) 3. Total 10 sessions: 3.1 mL (0.1 mg/mL) and 1.6 mL (0.175 mg/mL)	1. VAS for pain	1. Improved (10 to 3)	n.r.	1. Electro-acupuncture
12	Oh [39] (2015)	Case report	*n* = 1	Lateral malleolus avulsion fracture patient with ankle pain	1. Concentration: 0.05 mg/mL or 0.1 mg/mL 2. 1 session: 0.2 mL 3. Total 9 sessions: 0.6 mL(0.05 mg/mL) and 1.2 mL (0.1 mg/mL)	1. VAS for pain 2. AHS 3. ROM of ankle	1. Improved (8 to 3) 2. Improved 3. Improved -Flexion: 0° to 20° -Extension: 0° to 10°	Itching, redness in 1 case	1. Acupuncture 2. Herbal medicine
13	Kim [40] (2016)	Case report	*n* = 1	Peroneal nerve palsy with foot drop patient with ankle pain	1. Concentration: 0.1 mg/mL 2. 1 session: n.r. 3. Total 15 sessions: n.r.	1. NRS for pain 2. ROM of ankle (1) Sitting (2) Standing	1. Improved (10 to 0) 2. (1) Improved (0 to 30°) (2) Improved (0 to 30°)	n.r.	1. Acupuncture 2. Herbal medicine 3. Moxibustion 4. Cupping 5. Physical therapy
14	Oh [41] (2017)	Case report	*n* = 1	Acute inflammatory arthritis of ankle joint patient with ankle pain	1. Concentration: 0.05 mg/mL 2. 1 session: 0.3 mL 3. Total 9 sessions: 2.7 mL	1. NRS for pain 2. Blood test (1) CRP (2) ESR 4. ROM of ankle 5. Swelling	1. Improved (10 to 2) 2. (1) Improved (Positive to negative) (2) Improved (45 to 23) 4. Not improved 5. Improved (0.7 cm decrease)	None	1. Acupuncture 2. Warm acupuncture 3. Herbal medicine 4. Moxibustion

* For case reports or single-arm studies (*n* = 1), the term *“improved”* was used when authors reported post-treatment improvements in clinical outcomes, although no statistical analysis was conducted. For RCTs and CCTs, statistical significance (e.g., *p*-values) was reported when available. AHS: American Orthopaedic Foot & Ankle Society Hindfoot Score; BVA: bee venom acupuncture; CCT: case-controlled clinical trial; CRP: C-reactive protein; ESR: erythrocyte sedimentation rate; IR: infrared radiation; n.r.: not reported; NRS: numeral rating scale; NS: no significant difference between groups or before/after intervention; RCT: randomized controlled trial; ROM: range of motion; VAS: visual analog scale.

**Table 2 toxins-17-00257-t002:** Distribution of studies and patient populations by medical condition.

Medical Conditions of Participants	Number of Studies *n* (%)	Number of Patients Total (Range)
Traumatic conditions (ankle sprain, traumatic partial tear, and malleolus avulsion fracture)	6 (42.9%)	73 (1–18)
Inflammatory conditions (synovitis of ankle joint with osteonecrosis, rheumatoid arthritis, and acute inflammatory arthritis)	3 (21.4%)	3 (1)
Neuropathic conditions (tarsal tunnel syndrome and peroneal nerve palsy with foot drop)	2 (14.3%)	2 (1)
Other conditions (post-operative and anterior impingement syndrome)	2 (14.3%)	2 (1)

One study [28] did not report the medical conditions that may have caused ankle pain and was therefore excluded from the classification.

**Table 3 toxins-17-00257-t003:** Concentrations and volumes of bee venom acupuncture administered, categorized by the specific medical conditions of the patients.

Medical Conditions of Participants	Concentration * (mg/mL)	Volume	
		Volume Per 1 Session (mL)	Volume for Entire Treatment (mL)
Traumatic conditions (ankle sprain, traumatic partial tear, and malleolus avulsion fracture)	0.05–0.33 (3000:1–20,000:1)	0.04–2.0	0.28–8.0
Inflammatory conditions (synovitis of ankle joint with osteonecrosis, rheumatoid arthritis, and acute inflammatory arthritis)	0.05 (20,000:1)	0.1–2.5	2.7–16.0
Neuropathic conditions (tarsal tunnel syndrome and peroneal nerve palsy with foot drop)	0.1–0.175 (5700:1–10,000:1)	0.1–0.6	1.6–3.1
Other conditions (post-operative and anterior impingement syndrome)	0.1–0.5 (2000:1–10,000:1)	0.06–0.4	0.2–1.6

* A:1 is the dilution rate of A amount of normal saline with 1 g of bee venom (example: 20,000:1 means the bee venom acupuncture concentration of 20,000 mL of normal saline with 1 g of bee venom dilution).

**Table 4 toxins-17-00257-t004:** Risk-of-bias assessment.

First Author, Year	Selection Bias		Performance Bias	Detection Bias	Attrition Bias	Reporting Bias
	Random Sequence Generation	Allocation Concealment	Blinding of Participants and Personnel	Blinding of Outcome Assessment	Incomplete Outcome Data	Selective Reporting
Song, 2005 [31]	L	L	L	U	L	U
Seo, 2006 [34]	L	L	H	U	H	U
Kang, 2008 [36]	L	L	H	U	H	U

H: high risk; L: low risk; U: unclear risk.

## Data Availability

No new data were created or analyzed in this study.

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
