# Peer review of "Clinical Evidence of Bee Venom Acupuncture for Ankle Pain: A Review of Clinical Research"

_toxins, 2025, doi:10.3390/toxins17050257_

Round 1

Reviewer 1 Report

Comments and Suggestions for Authors
  • The search method works, but it doesn't narrow the languages used or give you any way to check the quality of these studies.
  • A graphical abstract is strongly recommended.
  • Language: While the overall English language quality is acceptable, minor revisions are needed to improve clarity and readability.

Author Response

 We appreciate the time given and efforts made by the editor and referees in reviewing this paper. Please see the attachment.

Reviewer 2 Report

Comments and Suggestions for Authors

Comments:

Authors aimed to investigate the effectiveness and safety of bee venom acupuncture (BVA) in the treatment of ankle pain including clinical studies that utilized BVA for the treatment of ankle pain and studies that included pain-related assessment tools. For this, ten electronic databases were searched for articles published up to March 2025. The review indicates that BVA may be helpful in controlling ankle pain and improving function in cases of ankle pain caused by traumatic, inflammatory, and neuropathic disorders. This study has several limitations. (i) First, among the 15 studies, only 3 were high-quality randomized controlled trials (RCTs), prospective studies that strictly control clinical research environments to measure and compare the therapeutic effects between experimental and control groups; (ii) Secondly, all included studies were conducted in South Korea and (iii) Thirdly, only half of the 14 studies reported adverse and none reported serious adverse events; however, the evidence remains limited. As positive points to be highlighted in this review, we can mention the author´s recognition that (i) future studies should focus on conducting high-quality RCTs, and systematic reviews and meta-analyses based on RCTs should also be performed; (ii) the possibility of publication bias, and careful interpretation of the results obtained in South Corea is needed and future multinational, multicenter RCTs on BVA for ankle pain including report on adverse effects.are necessary to verify the efficacy and safety of BVA.   

Conclusions are consistent with the data presented by the analyzed studies. Tables and figures are apropriate.

However, primary errors regarding references were made during the preparation of the manuscript.

Comments on REFERENCES

  • Some notes involving mistakes on references cited throughout the manuscript:

Line 42 ....Bee venom (BV) is an animal venom secreted from the venom glands of bees, containing peptides (e.g., melittin, apamin, and adolapin), enzymes (e.g., phospholipase A2 and hyaluronidase), amino acids, and volatile compounds [11, 12].

Comment: This ref 11 is about Apitoxin and do not correspond to this sentence. Please correct

In line 44.....In animal studies, BV has been shown to have anti-inflammatory, anti-nociceptive, anti-cancer, and anti-allergic effects [3-5].

Comment: Reference 3 estimates on the prevalence of musculoskeletal pain of five different anatomical areas and ten anatomical sites, and their consequences and risk groups in the general Dutch population; Reference 4  is about Prevalence of foot and ankle conditions in a multiethnic community sample of older adults and Reference 5 is about Posttraumatic ankle arthritis.

These references do not correspond to this sentence. Please correct

In line 48.....In some East Asian countries, traditional clinics use bee venom acupuncture (BVA) for disease treatment [6]. Comment: Reference 6 is about Ankle sprains in athletes and do not correspond to this sentence. Please correct

Additional comment: References must be numbered in order of appearance in the text . Serious errors in relation to references 13 to 17 in the manuscript. These references are missing in the manuscrit. Please see lines 44 until 49. In the item REFERENCES, references number 11, 12, 13, 15 and 16 are not present in the manuscript.

 (2) Please check carefully all described references in the item References

  • According to Instructions for Authors, Journal Articles:
    1. Author 1, A.B.; Author 2, C.D. Title of the article. Abbreviated Journal Name Year, Volume, page range.

Author Response

(The authors gave the same response as above.)

Reviewer 3 Report

Comments and Suggestions for Authors

This review examines a potential application of bee venom, compiling diverse studies to provide an overview of the current landscape. However, the manuscript lacks a critical analysis of the findings and fails to adequately address its primary objective: investigating the effectiveness and safety of bee venom. To strengthen the study, the authors should reconsider the aims or establish clear parameters that enable definitive conclusions. As it stands, the conclusion acknowledges that "it is difficult to draw definitive conclusions".

  1. Please avoid repeating words in the same sentence—for example, clinical in the title. Lines 5-6: injuries, lines 6-7: ankle, lines 28-29: pain, line 42: venom and so on. Please check the entire manuscript.
  2. Line 14-15. I do not understand the term dosage here, as the units are not compatible.
  3. The list of keywords must be different from those used in the title.
  4. The authors must revise the literature and clarify the relevance of this new review. For example, many studies addressed the same topic: https://www.sciencedirect.com/science/article/pii/S0041010118303933, https://pmc.ncbi.nlm.nih.gov/articles/PMC1062163/, among others. The authors can use a figure or table to summarise the state-of-the-art and highlight the gaps and points addressed in this novel manuscript.
  5. Line 42. 2 should be subscript.
  6. The authors can include more details of venom composition. Are all bee venoms the same? What is the composition of the most used bee venoms? What is the abundance of those components?
  7. What are the main clinical manifestations of bee envenomation? In some cases, antivenom is required to treat bee envenomation.
  8. Lines 50. What are the existing studies? References are important.
  9. Figure legends. Please include a brief description in addition to the title.
  10. Figure 2 can be elaborated using more professional software. Excel is not the best one for scientific purposes. The authors can consider using heatmap. I do not understand why the number were included in the middle of the figure, if y-axis has been provided.
  11. Please add a brief description in addition to the title.
  12. Table 1. I do not understand the countries in the first column.
  13. Table 1 presents unclear data regarding concentration and dosage. In many instances, the authors provide volumes which do not correspond directly to either concentration or dosage.
  14. The column main result is not clear. The authors must explain the parameters in the legend and also standardise the presentation. In some cases, the words improved appear and other statistical results without a clear definition of the statistical test employed.
  15. How is the concentration of bee venom estimated? Please include these details in the manuscript.
  16. Figures 3-4. Numbers are not necessary as they can clearly be identified from the y-axis.
  17. Please discuss the potential causes of adverse events.
  18. Lies174-180. The authors explored the therapeutic side of melittin. However, it is a highly toxic and haemolytic peptide. It is used as a positive control in assays assessing the impact of peptides on red blood cells' integrity.
  19. Lines 180. particular.
  20. The discussion is poor and must be rewritten. I was expecting a critical discussion and view of all interesting findings presented after the extensive literature review. Gaps, failures, novel directions and biases were not properly addressed in this section.
  21. The positive outcomes have been attributed only to melittin. However, what about the potential contributions of other components? Why was the focus not placed solely on melittin? Additionally, the source of the bee venom has not been specified, raising the question of whether all bee venoms exhibit the same composition and effects.
  22. The manuscript would benefit from the inclusion of a figure illustrating venom components and their associated pathways in pain treatment.
  23. The manuscript did not answer its main scientific question: investigate the effectiveness and safety of bee venom acupuncture. The authors must reformulate the aim or better address this point. 
  24. The quality of graphics must be improved. Excel does not provide high-quality images. 

Author Response

(The authors gave the same response as above.)

Round 2

Reviewer 3 Report

Comments and Suggestions for Authors

The authors have addressed most of the points raised by the reviewers. But, in my opinion, the lack of clear criteria for assessing the clinical effectiveness and safety does not allow the authors to answer the main question of this review. The authors need to define the parameters in the methodology section. If they do not have established them, how can they answer the aim of this review? If the parameters are not enough to draw the conclusions, the authors must rethink the aim of this manuscript. 

Author Response

 We appreciate the time given and efforts made by the editor and referees in reviewing this 
paper. Pleases see the attachment.

Round 3

Reviewer 3 Report

Comments and Suggestions for Authors

The authors have included the criteria for assessing the paper's objective in the methodology section. I do not have further comments. I recommend the publication in the current form.